# Impact of COVID-19 Restriction on Weight, Physical Activity, Diet and Psychological Distress on People with PCOS

**DOI:** 10.3390/nu15112579

**Published:** 2023-05-31

**Authors:** Margaret McGowan, Siew Lim, Sharleen L. O’Reilly, Cheryce L. Harrison, Joanne Enticott, Helena Teede, Stephanie Cowan, Lisa J. Moran

**Affiliations:** 1Monash Centre Health Research and Implementation (MCHRI), Clayton 3168, Australia; 2Eastern Health Clinical School, Monash University, Box Hill 3128, Australia; 3UCD Institute of Food and Health, School of Agriculture and Food Science, University College Dublin, Belfield, D04 V1W8 Dublin, Ireland; 4Diabetes Unit Monash Health, Clayton 3168, Australia

**Keywords:** polycystic ovary syndrome, COVID-19, weight, physical activity, diet, psychological distress

## Abstract

Background: People with polycystic ovary syndrome (PCOS) have higher weight gain and psychological distress compared to those without PCOS. While COVID-19 restrictions led to population level adverse changes in lifestyle, weight gain and psychological distress, their impact on people with PCOS is unclear. The aim of this study was to investigate the impact the 2020 COVID-19 restrictions had on weight, physical activity, diet and psychological distress for Australians with PCOS. Methods: Australian reproductive-aged women participated in an online survey with assessment of weight, physical activity, diet and psychological distress. Multivariable logistic and linear regression were used to examine associations between PCOS and residential location with health outcomes. Results: On adjusted analysis, those with PCOS gained more weight (2.9%; 95% CI; 0.027–3.020; *p* = 0.046), were less likely to meet physical activity recommendations (OR 0.50; 95% CI; 0.32–0.79; *p* = 0.003) and had higher sugar-sweetened beverage intake (OR 1.74; 95% CI 1.10–2.75; *p* = 0.019) but no differences in psychological distress compared to women without PCOS. Conclusions: People with PCOS were more adversely affected by COVID-19 restrictions, which may worsen their clinical features and disease burden. Additional health care support may be necessary to assist people with PCOS to meet dietary and physical activity recommendations.

## 1. Introduction

COVID-19 forced many major cities across the world into lockdowns at the start of 2020 [1]. In Australia, the level of restrictions and advice given varied between states and regions [2]. The general advice given by the federal government in 2020 was those who could work from home were told to work from home [2]. During lockdowns or stay-at-home orders, Australian’s could only leave their house for four reasons; essential shopping, to seek medical advice, to work or to provide care [2]. The COVID-19 pandemic and the subsequent lockdowns impacted Australians’ physical and psychological health [3]. Worldwide, COVID-19 restrictions and/or lockdowns were associated with weight gain [4,5], reductions in passive physical activity [6] and significant impacts on women’s overall physical activity levels [7,8] as well as suboptimal dietary changes with an increased number of meals/day, snacks/day and frequency of discretionary food consumption [9,10,11]. COVID-19 restrictions were also associated with worsened psychological health [12]. In addition, these adverse changes in diet quality [13,14] and physical activity [15,16] have also been associated with increased psychological distress either before or during the pandemic. 

Preliminary evidence suggests that women with complex health issues, including polycystic ovary syndrome (PCOS), may be disproportionately affected by the COVID-19 restrictions [17]. PCOS is a cardiometabolic disorder which affects 8% to 13% of women [18]. In adults, it is diagnosed according to the International Guideline updated Rotterdam criteria requiring two of the three features of oligo-anovulation (reduced ovulation) or irregular menstrual cycles; biochemical hyperandrogenism (elevated circulating male hormones or androgens such as testosterone) or clinical hyperandrogenism (effects of androgens on body tissues including hirsutism or excess hair growth); and presence of polycystic ovaries on ultrasound [19]. It is associated with variable reproductive (infertility), metabolic (insulin resistance and increased risk for type 2 diabetes (T2DM) and cardiovascular disease) and psychological features (increased prevalence of anxiety, depression [19,20] and disordered eating/eating disorders [21]). People with PCOS also have a higher prevalence of weight gain and obesity [22,23] which can further worsen clinical features [24]. In keeping with this, the international evidence-based guidelines for PCOS recommend weight management (prevention of weight gain or achieving and maintaining modest weight loss) through lifestyle interventions (diet, physical activity, and behavioural), as first-line therapy [19]. 

Limited research has explored the impact of COVID-19 restrictions on those with PCOS. While some reports disruptions to weight management support [25] and worsened levels of stress, anxiety and depression [25,26,27], other research suggests no significant differences in weight gain compared to those without PCOS [26]. Given obesity worsens the features of PCOS [24], adverse changes in lifestyle and weight reported in the general population during COVID-19 [4,5] could negatively influence the health of those with PCOS. However, the full impact of COVID-19 restrictions on those with PCOS are currently unknown. This highlights the need for further research to understand the physical and mental health impact COVID-19 restrictions had on those with PCOS to inform health care support strategies for ongoing healthy lifestyle management. 

The primary aim of this study was to investigate the impact the 2020 COVID-19 restrictions had on Australians with and without PCOS, specifically weight, physical activity, diet and psychological distress. The secondary aim was to investigate the impact greater restrictions had on weight, physical activity, diet and psychological distress in women with and without PCOS. It is hypothesised that those with PCOS had greater weight gain, lower physical activity, increased intake of unhealthy foods and higher psychological distress compared to women without PCOS. 

## 2. Materials and Methods

### 2.1. Study Design and Population

A cross-sectional nationwide online survey was conducted from 15 October to 7 November 2020 [28]. The survey explored weight, physical activity, diet and psychological distress in relation to sociodemographic information in Australia during the COVID-19 pandemic in 2020 in women of reproductive age (18–50 years). We understand that not all those living with PCOS identify as women, thus have tried, where possible, to refer to those with PCOS in gender neutral terms to foster inclusivity. Women were contacted via email by external cross-panel market research providers using a well-established database and reimbursement in accordance with ISO 26362 [29] and industry requirements. Following online informed consent, participants anonymously completed a 10-min survey consisting of short answers and multiple-choice questions. To ensure a broad representation (ages, state/territory and rurality) within the study cohort, responses were examined after 4 days and further targeted sampling was conducted. Once 1000 responses were received the survey was concluded. The methodology has been described in detail previously [28]. 

### 2.2. Survey Variables Population Characteristics

The survey included multiple-choice questions to assess participants’ age group, cultural or ethnic group, highest level of education completed, employment status before the pandemic, changes in employment status during the 2020 portion of the pandemic and annual household income before tax. Employment status was categorised as full-time, part-time/casual, student or other, which included homemakers, retirees, and those on government assistance (family payments, Job Keeper, Job Seeker or disability support). Urban or rural/remote location and state of residence was determined based on postcode. Education was divided into 3 categories: university (completion of an undergraduate degree or higher), TAFE/certifications (completion of any TAFE qualification or certificate) or high school education or lower. TAFE (technical and future education), also known as vocational education, provides a range of technical, vocation and training courses such as certificate levels I, II, III and IV. Participants were asked to best describe their culture based on 4 categories: European (British, Irish, Western European, Northern European, Southern European, Southeast European, Eastern European, European or North American), Oceania (Australian peoples, New Zealand peoples, Pacific Islanders), Asian (North East Asian, Southern and Central Asian, or South East Asian) and Other (Arab, Jewish, Peoples of Sudan, North African, Middle East, Sothern African or West African). The survey also included questions on the number of children living in the household, dietary intake, alcohol consumption, physical activity and psychological distress. Data on self-reported weight and height were used to calculate weight change, percent weight change and body mass index (kg/m^2^, BMI), which was categorised as underweight (<18.5 kg/m^2^), normal weight (18.5–24.9 kg/m^2^), overweight (25.0–29.9 kg/m^2^) and obese (>30.0 kg/m^2^) [30]; BMI was dichotomised into underweight and healthy weight (≤24.9 kg/m^2^) or having overweight or obesity (>24.9 kg/m^2^). Reported weight prior to the COVID-19 restrictions was referred to as “pre-pandemic weight” and the weight reported at the time of the survey was referred to as “current weight”. PCOS status was self-reported through participants being asked “Have you ever had, been diagnosed or treated for PCOS?”. 

### 2.3. Level of Viral Transmission Rates and Corresponding Lockdown Restrictions

From mid-March 2020, the Australian government took several measures to increase physical distancing and reduce human contact to reduce the transmission of COVID-19; the restrictions changed significantly throughout the course of 2020 and varied between states [2]. The Australian Government introduced restrictions to international travel from all countries as well as travel between states, banning of large gatherings and closure of non-essential shops, entertainment, food (excluding takeaway) and recreation venues to reduce transmission of the virus. By the end of March, all Australian states and territories were under stay-at-home orders, and people were only allowed to leave their homes for four reasons: food and supplies, medical care, physical activity and work or education (Appendix A) [2]. Each state determined their own specific stay-at-home order rules. In mid-May, community transmission was very low and most national and state restrictions were eased across the country (Appendix A) [2]. From this time, no further major restrictions were introduced across Australia in 2020, with the exception of metropolitan Melbourne [2]. Stay-at-home orders were re-established in June for some areas of Melbourne, with all of metropolitan Melbourne under a stay-at-home order between August to October (Appendix A) to contain the virus. This study was conducted at the end of the lockdown in metropolitan Melbourne when restrictions started to ease (15 October and 7 November 2020). Prior to the survey, metropolitan Melbourne residents spent 119–153 days under stay-at-home orders compared to 32–49 for the rest of Australia [2]. Each participants’ postcode and state was used to determine residential location and dichotomised as exposure to high viral transmission rates and strict lockdown restrictions (i.e., women living in metropolitan Melbourne) vs. exposure to low viral transmission rates and less strict lockdown restrictions (i.e., women living in any other region in Australia). 

### 2.4. Physical Activity

The Active Australia Survey, a validated questionnaire on physical activity and sedentary behaviour, was used to assess frequency of physical activity [31]. Women were asked to report the frequency and total minutes spent in the last week on walking briskly, moderate and vigorous leisure activities, and vigorous household or garden chores. Physical activity outliers were identified by summing total moderate leisure and vigorous leisure frequencies and marked as missing if the sum was >56 occasions [32]. Total metabolic minutes per week (MET.min/week) was calculated by summing the products of each type of physical activity with its metabolic equivalent value (MET.min/week = (weekly walking minutes × 3.33) + (weekly moderate leisure minutes × 3.33) + (weekly vigorous leisure × 6.66) + (weekly vigorous chores × 6.66)) [33]. Total MET.min/week were categorised into none or very low (<33.3 MET.min/week), low (33.3–500 MET.min/week), moderate (500–1000 MET.min/week), or high level of physical activity (≥1000 MET.min/week) categories [32]. Physical activity was categorised into meeting (≥500 MET.min/week) and not meeting (<500 MET.min/week) Australia’s Physical Activity Guidelines [32].

### 2.5. Dietary Intake

Dietary survey questions were developed based on the Irish National COVID-19 Food Survey and adapted to the Australian settings using the Australian Guide to Healthy Eating to collect information on the consumption of sugar-sweetened beverages (SSB), alcohol and total discretionary foods. Frequency of discretionary food intake was obtained through multiple choice questions of 1–2 times/day, 3–5 times/day, 5–7 times/day, 7–10 times/day, >10 times/day, do not eat every day, never and I do not know. Frequency of discretionary foods were then binary coded into ≤2 times/day and ≥3 times/day to align with the Australian Guide to Healthy Eating discretionary food recommendations for women of <2.5 times a day [34,35]. Frequency of SSB intake was obtained using multiple-choice options: <1 time/week, 1–3 times/week, 4–6 times/week, ≥once/day, never and I do not know, which were then binary coded into less than weekly and most days/daily based on the median intake. Frequency of alcohol consumption was obtained through multiple-choice options of 2–3 times a week, 2–4 times per week, ≥4 times a month, monthly or less, never and I do not know/prefer not to answer, and collapsed into low frequency (never and less than monthly) and high (2–3 times a week, 2–4 times per week or ≥4 times a month). Responses of “I do not know”, or “I prefer not to answer” were excluded from binomial logistic regression. Fruit and vegetables serving number were self-reported (as continuous variables) with outliers deemed as values greater than 3 times the 75th percentile, as per Yaroch et al. [36].

### 2.6. Psychological Distress

The K10, developed by Kessler et al., is a validated short dimensional assessment tool consisting of 10 questions on anxiety (4 questions) and depression (6 questions), used for measuring non-specific psychological distress [37,38]. All questions are weighted evenly (1–5 points) with a score total between 10 and 50 [38]. The K10 was validated in 2000 for use at a population level in the US National Health Interview Survey [38] and has been used by the Australian Bureau of Statistics (ABS) and NSW mental health services to measure psychological distress since 2001 [37]. Score classification were as follows: 10–15 low, 16–21 moderate, 22–29 high and 30–35 very high, with the classifications being collapsed further into low/moderate (10–21) and high/very high (22–35) [38].

### 2.7. Statistical Analysis

Significance was set at *p* < 0.05. These analyses used the statistical software package IBM SPSS for Windows Version 26 (SPSS INC., Chicago, IL, USA) and Stata Version 15.1 (StataCorp, College Station, TX, USA). Chi-squared tests were used to compare demographic, BMI, direction of weight change (classified as weight loss/no change or weight gain), physical activity, SSB, discretionary foods, alcohol and psychological distress between those with versus without PCOS. Mann–Whitney tests were used to compare continuous variables that are not normally distributed (e.g., K10). Independent samples t-tests were used to compare normally distributed variables such as BMI, weight change and weight change percent. Multivariable logistic and linear regression were used to examine associations of the dependent variables weight change, physical activity, dietary intake and psychological distress with PCOS status. To explore differences in the association by level of lockdown restrictions, an interaction term was added to the model (PCOS × level of restrictions) and stratified analyses were conducted if the interaction term *p*-value was less than 0.05. All models were adjusted for predetermined factors known to be associated with weight management, psychological distress, physical activity and dietary intake including age group, location (in non-stratified analyses), annual household income before tax, ethnicity, highest level of education completed, employment (at time of survey) and number of children in the household. 

## 3. Results

A total of 1005 women responded to the survey, with 113 (11.2%) self-identified as having PCOS and 892 women who did not have PCOS. The majority of women were aged between 25 and 34 years old (34.7%) and one in four women lived in metropolitan Melbourne, Victoria. Table 1 outlines the demographic characteristics of all respondents. There were no statistically significant demographic differences between those with and without PCOS. 

When compared to women without PCOS, those with PCOS had a significantly higher pre-pandemic weight (*p* = 0.008) and had higher weight and BMI when the survey was conducted (*p* < 0.001 and *p* < 0.001, respectively) (Table 2). Between March 2020 and when the survey was completed (October/November 2020), 41.1% of women gained weight, without a significant difference between total (1.8 vs. 1.1 kg, *p* = 0.911) or percent (2.9 vs. 1.6%, *p* = 0.061) weight gain for those with and without PCOS. On adjusted analysis, those with PCOS gained significantly larger percent weight compared to women without PCOS (1.61%; 95% CI; 0.027–3.020; *p* = 0.046) (Table 2). TAFE/certificate education level compared to university degree and part time, full time and other employment (current) compared to university students were positively associated with weight gain (Appendix A).

Physical activity for people with and woman without PCOS is reported in Figure 1. People with PCOS (42.5%) were significantly less likely to meet the physical activity recommendations compared to women without PCOS (59.3%) on both unadjusted (*p* ≤ 0.001) and adjusted analyses (OR; 0.50, 95% CI; 0.32–0.79; *p* = 0.003). The proportion of women in each physical activity category (no/little, low, moderate and high) was significantly different between those with and without PCOS (*p* = 0.007) (Figure 1), with those with PCOS having a pattern of more low, very little or no physical activity and less high or moderate time spent physically active. On multivariable analysis, the factors associated with not meeting physical activity recommendation include identifying as Asian, lower levels of education (TAFE/Certification and high school) and an increase in age (across all categories) (Appendix A).

Across the entire study sample, women reported a mean intake of fruit and vegetable of 1.75 and 2.58 servings respectively, around half reported a low SSB and alcohol intake and over two thirds reported a low discretionary foods intake. There were no differences in dietary intake for people with and without PCOS on unadjusted analysis (Table 3). On adjusted analysis, those with PCOS were more likely to consume sugar sweetened beverage at a high frequency (OR 1.74; 95% CI 1.10–2.75; *p* = 0.019) (Table 3). On multivariable analysis, income had a negative association, while identifying culturally as Arab, Jewish, Peoples of Sudan, North African, Middle East, Sothern African or West African and having 3+ children had a positive association with vegetable consumption. Being European was associated with higher frequency of SSB consumption and being aged between 25–34 years, and having a highest level of education of TAFE/certification and having 2 children were associated with high frequency of discretionary food consumption (Appendix A). 

Most women (59.5%) reported levels of low or moderate psychological distress with no differences between those with and women without PCOS (Table 4). On adjusted analysis, women living in metropolitan Melbourne had a significantly higher K10 score (2.02; 95% CI 0.62–3.42; *p* = 0.005) and were more likely to have a higher category (low/moderate or high/very high) of psychological distress (OR 1.61; 95% CI 1.13–2.29; *p* = 0.008) (Appendix A). An increase in age and employment categorised as other (homemakers, retirees and those on government assistance) were both associated with higher psychological distress (Appendix A).

There was no significant interaction between PCOS diagnosis and residential location for any variable. 

## 4. Discussion

The COVID-19 pandemic had a significant impact on mental and physical health at a global scale [3]. We report that those with PCOS gained more weight, were less likely to meet physical activity recommendation and had higher intake of SSB compared to women without PCOS. 

Weight gain is a recognised outcome of COVID-19 restrictions and lockdowns worldwide [4], with ~30% of the population gaining weight [4] and meta-analyses reporting an overall average 0.93 kg weight gain [5]. This is broadly consistent with our findings of 40.6% of women gaining weight and a 1.1 kg weight gain for those without PCOS. Furthermore, we report a similar proportion of women with PCOS gained weight (45%) compared to prior research (48.5%) [26]. Pre-pandemic, community-based population studies reported higher annual weight gain for women with (0.81 kg, 1.2%) compared to those without (0.55 kg, 0.94%) PCOS (*p* < 0.001) [39]. Here, in our community cohort of similar demographic characteristics, those with and without PCOS doubled their non-pandemic weight gain (1.8 vs. 1.1 kg) with a higher adjusted percent weight gain for those with versus without PCOS (2.9 vs. 1.6%). We note this difference in weight gain is much higher in magnitude in our current study (i.e., 1.8 fold vs. 1.3 fold for % weight gain) and over a shorter period (7–8 vs. 12 months) compared to Awoke et al. (12 months) [39], suggesting that the higher weight gain observed in general in PCOS was compounded here by the adverse pandemic conditions. However, as we are lacking pre-pandemic weight change data in this study it is not possible to confirm this here. However, as weight gain [24] and obesity [24] worsen the reproductive, metabolic and psychological features of PCOS [24], the greater weight gain observed here for those with PCOS would likely further exacerbate their clinical presentation. Our findings therefore highlight that those with PCOS are potentially left with greater health disparities compared to the general population after COVID-19 restrictions in the context of their higher weight gain. All people in the current post-lockdown world will need ongoing support with lifestyle and weight management. However, those with PCOS should arguably have access to additional support compared with pre-pandemic levels [22,23] given the recognised association of excess adiposity with health outcomes and first-line role of lifestyle interventions in international PCOS evidence-based guidelines [19].

We report a lesser proportion of people with PCOS met the physical activity guidelines (42.5 vs. 59.3%) which is consistent with a meta-analyses reporting those with PCOS perform less physical activity than women without PCOS [40]. It is possible that physical activity levels in those with PCOS was lower pre-pandemic compared to those without PCOS, with this difference being continued during the pandemic. Furthermore, the reduced levels of physical activity in PCOS could have contributed to their significantly higher weight gain as those with PCOS are known to have greater longitudinal weight gain [23], with potential resultant impacts on other metabolic, reproductive and psychological features of PCOS [24]. In the current post-pandemic world, women’s physical activity levels may still be impacted by COVID-19 restrictions and disease transmission risk. This is supported by one study reporting women were more hesitant to return to their pre-pandemic routines compared to men [41]. Focus should be given on the reintegration of exercise in the current post-COVID-19 pandemic world to assist those with PCOS to meet physical activity recommendations and to aid with achieving lifestyle management recommendations consistent with evidence-based guidelines [19]. We also observed higher SSB intake on adjusted analysis for those with compared to women without PCOS. This is consistent with some, albeit inconsistent, reports of higher discretionary food intake for women with PCOS in non-pandemic periods. This higher intake is of key importance here as SSBs are associated not only with increased risk of T2DM, obesity, coronary heart disease and stroke [42] as conditions with a higher prevalence for those with PCOS [19,20,21,22,23,43], but also weight gain (with one study reporting a 0.22 kg weight gain over 1 year for each SSB serving consumed daily [44]). This highlights their potential contribution to the higher weight gain observed here for those with PCOS. 

Periods of COVID-19 lockdowns resulted in increased levels of stress, depression and anxiety across the entire population [12]. The influence of COVID-19 restrictions on psychological wellbeing is multifactorial and may relate to factors including fear of infection, inadequate and misinformation, economic uncertainties and government restrictions resulting in social isolations, frustration and boredom [45,46]. As those with PCOS have increased levels of stress, anxiety and depression compared to the general population prior to the pandemic [20,47], we hypothesised even greater disparities arising from the 2020 lockdowns. However, this was not reflected in our data, with similar K10 scores for those with and women without PCOS and moderate psychological distress reported by all women. In comparison, a UK study in women with PCOS reported extremely severe depression, anxiety and stress [27], assessed with the DASS [48]. Our lower levels of psychological distress may be due to the K10 only measuring general psychological distress and not individually measuring anxiety, depression and stress symptoms [48]. It could also be related to the differing situations in Australia and UK during the 2020 pandemic restrictions, with higher rates of disease transmission in UK [49] which negatively impacted mental health [45]. Alternatively, psychological distress in PCOS may have been less adversely affected by COVID-19 restrictions, with some women reporting reduced anxiety and stress as they experienced less social stigma and judgement about excess hair growth and weight due to their reduced socialisation as a result of restrictions [25]. While we reported no differential effect of lockdown severity on psychological distress for people with and without PCOS, all women who lived through a prolonged lockdown (metropolitan Melbourne) had higher levels of distress compared to those in a shorter lockdown. This is consistent with quarantine duration, with a longer duration experienced by those residing in metropolitan Melbourne [2,50,51] being associated with poorer mental health outcomes [45]. Given the likely worsening in psychological health reported in other populations over COVID-19 [52], and with the psychological impact of quarantines seen from 6 months to 3 years after their cessation [53,54], the importance of psychological support now after the pandemic is highlighted for all women. This is of particular importance for women with PCOS living in metropolitan Melbourne given their existing high levels of depression, stress and anxiety [20,47] and the limited post-pandemic access to mental health care services, with one in three psychologists in Australia being unable to take on new clients, compared to one in hundred pre-pandemic [55].

We report methodological strengths here of the large study size (*n* = 1005) and the PCOS subpopulation being reflective of the community prevalence of 8–13% [18], likely giving an accurate representation of both populations. The cross-sectional nature of this paper does not allow for causation to be inferred and may not accurately capture the changes that occurred throughout the 2020 period of the pandemic. Thus, we are unable to determine a causal relationship of COVID-19 restrictions resulting in greater weight gain in those with PCOS as they were already more likely to gain more weight under standard circumstances. Although the questionnaires used were validated, with the exception of the dietary survey, they heavily relied on self-reported data, including PCOS diagnosis, weight, exercise and diet. While this may be a limiting factor, the use of self-report data allowed a large population to be surveyed across Australia during physical distancing restrictions. Self-reported PCOS status has also previously been reported to be strongly correlated with key diagnostic symptoms of PCOS [23]. Self-reported weight and BMI are likely underestimated while height is likely overestimated [56], thus the full magnitude of weight change throughout 2020 may not be fully understood. Further, as the entire survey did not have to be completed, the potential for reporting bias should be acknowledged. Lastly, the study has considered several confounding factors that may influence the outcomes measured; however, the factors are limited to the survey conducted and did not include sleep and medications, two factors which may result in weight changes.

In conclusion, we found people with PCOS reported gaining more weight, lower physical activity and higher SSB consumption compared to women without PCOS throughout the 2020 COVID-19 pandemic period. Additional health care support is required to assist people with PCOS to meet recommendations for optimising diet, physical activity and weight to reduce the health burden and long-term impacts of COVID-19 restrictions. This could include additional assistance in reducing barriers such as access and affordability of care, as well as empathy and understanding from health care providers. Future research should examine the impact additional lockdowns in Australia over 2021 had on those with and without PCOS to observe ongoing trends and to determine if additional resources and support are needed to offset the effects of restrictions. In a clinical setting it is important to acknowledge the impact the COVID-19 restrictions and lockdowns had on all women, especially those with PCOS seeking lifestyle changes and weight management. This highlights the need to support those with vulnerability to the adverse effects of lockdown, including those with PCOS.

## Figures and Tables

**Figure 1 nutrients-15-02579-f001:**
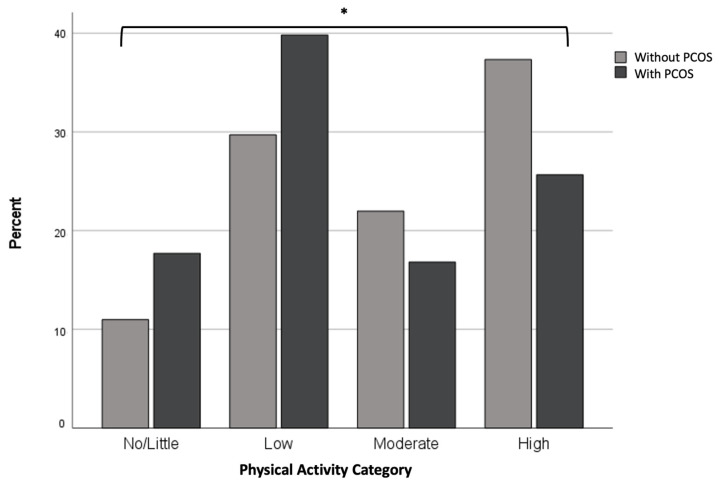
Physical activity for people with and people without PCOS. Data are presented as (%) and were analysed by chi-squared test. There is a significant different between the categorization (level) of physical activity between people with and without PCOS (* *p* = 0.007); PCOS: Polycystic Ovary Syndrome.

**Table 1 nutrients-15-02579-t001:** Demographic characteristics.

		All *n* (%)	With PCOS *n* (%)	Without PCOS *n* (%)	*p*
Age (years)	18–24	158 (15.7)	12 (10.6)	146 (16.4)	0.257
	25–34	349 (34.7)	36 (31.9)	313 (35.1)	
	35–44	329 (32.7)	43 (38.0)	286 (32.1)	
	>45	169 (16.8)	22 (19.5)	147 (16.5)	
Location	Metropolitan Melbourne	250 (24.9)	30 (26.5)	221 (24.7)	0.662
	Non-Metro Melbourne	755 (75.1)	83 (73.5)	673 (75.3)	
Income (AUD)	0–24,999	59 (6.8)	6 (6.1)	53 (6.9)	0.925
	25,000–49,000	91 (10.5)	10 (10.1)	81 (10.6)	
	50,000–124,999	410 (47.4)	51 (51.5)	359 (46.9)	
	125,000–149,999	94 (10.9)	9 (9.1)	85 (11.1)	
	>150,000	211 (24.4)	23 (23.2)	188 (24.5)	
Culture	Oceanian	561 (58.4)	69 (61.6)	492 (58.0)	0.472
	European	242 (25.2)	28 (25.0)	214 (25.2)	
	Asian	126 (13.1)	10 (8.9)	116 (13.7)	
	Other	31 (3.2)	5 (4.5)	26 (3.1)	
Education	University degree	198 (19.8)	15 (13.3)	183 (20.7)	0.128
	TAFE/Certificate	240 (24.0)	26 (23.0)	214 (24.2)	
	High school or lower	561 (56.2)	72 (63.7)	489 (55.2)	
Number of Children	0	573 (57.1)	61 (54.0)	512 (57.5)	0.802
	1	158 (15.7)	17 (15.0)	141 (15.8)	
	2	190 (18.9)	25 (22.1)	165 (18.5)	
	3+	83 (8.3)	10 (8.8)	73 (8.2)	
Job (Pre)	Student	67 (6.8)	8 (7.1)	59 (6.8)	0.672
	Casual or Part-time	257 (26.1)	24 (21.2)	233 (26.8)	
	Full-time	511 (51.9)	62 (54.9)	449 (51.6)	
	Other	152 (15.4)	19 (16.8)	133 (15.3)	
Job (Change)	Yes	252 (25.1)	31 (27.4)	221 (24.8)	0.539
	No	753 (74.9)	82 (72.6)	671 (75.2)	
Job (Post)	Student	68 (6.9)	3 (2.7)	65 (7.5)	0.231
	Casual or Part-time	254 (25.8)	27 (23.9)	227 (26.1)	
	Full-time	451 (45.9)	56 (49.6)	395 (45.4)	
	Other	210 (21.4)	27 (23.9)	183 (21.0)	

Data are presented as *n* (%) and were analysed by a chi-squared test. PCOS: polycystic ovary syndrome.

**Table 2 nutrients-15-02579-t002:** Weight and BMI in those with and without PCOS.

Variable	All	With PCOS	Without PCOS	*p* (Unadjusted)	*p* (Adjusted)
Pre-pandemic weight (kg)	68.3 (17.4)	73.1 (20.4)	67.7 (17.0)	0.008	-
Current weight (kg)	69.9 (18.3)	76.1 (21.4)	69.1 (17.7)	<0.001	-
Weight change (kg)	1.1 (4.7)	1.8 (5.8)	1.1 (4.5)	0.158	0.911
Weight change (%)	1.7 (6.4)	2.9 (8.7)	1.6 (6.1)	0.061	0.046
Weight change (gained)	397 (41.1)	49 (45.0)	348 (40.6)	0.296	0.694
Current BMI (kg/m^2^)	25.7 (6.3)	28.5 (7.9)	25.3 (5.9)	<0.001	-
BMI—Underweight	33 (4.6)	2 (2.4)	31 (4.9)	<0.001	-
BMI—Healthy weight	384 (53.6)	33 (39.3)	351 (55.5)		
BMI—Overweight	157 (21.9)	19 (22.6)	138 (21.8)		
BMI—Obese	142 (19.8)	30 (35.7)	112 (17.7)		
BMI—Healthy/underweight	417 (58.2)	35 (41.7)	382 (60.4)	<0.001	-
BMI—Obese/overweight	299 (41.8)	49 (58.3)	250 (39.6)		

Data are presented as mean (standard deviation) for continuous and number (percent) for categorical data. Data were analysed with an independent samples *t*-test for continuous normally distributed and a chi-squared test for categorical data. BMI: body mass index; PCOS: polycystic ovary syndrome.

**Table 3 nutrients-15-02579-t003:** Dietary intake in people with and without PCOS.

Food	All	With PCOS	Without PCOS	*p* (Unadjusted)	*p* (Adjusted)
Fruit (serves/day)	1.75 (0.97)	1.75 (1.0)	1.75 (1.0)	0.965	0.778
Vegetable (serves/day)	2.58 (1.39)	2.81 (1.5)	2.55 (1.4)	0.092	0.179
SSB (low)	521 (53.9)	48 (46.2)	473 (54.8)	0.094	0.019
SSB (high)	446 (46.1)	56 (53.9)	390 (45.2)		
Discretionary Foods (low)	836 (85.5)	99 (89.2)	737 (85.0)	0.239	0.235
Discretionary Foods (high)	142 (14.5)	12 (10.8)	130 (15.0)		
Alcohol (low)	502 (51.0)	64 (58.2)	438 (50.1)	0.108	0.125
Alcohol (high)	483 (49.0)	46 (41.8)	437 (49.9)		

Data are presented as mean (standard deviation) for continuous and number (percent) for categorical data and were analysed with an independent *t*-test for continuous normally distributed, a Mann–Whitney Test for continuous non-normally distributed, a chi-squared test for categorical data and multivariable linear and logistic analysis for continuous data. SSB: sugar sweetened beverage; PCOS: polycystic ovary syndrome.

**Table 4 nutrients-15-02579-t004:** Psychological distress levels in people with and those without PCOS.

Psychological Distress	All	With PCOS	Without PCOS	*p* (Unadjusted)	*p* (Adjusted)
Mean K10	19 (13)	20 (11)	19 (13)	0.375	0.834
Low K10	301 (30.7)	28 (25.7)	273 (31.2)	0.584	-
Moderate K10	284 (28.9)	36 (33.0)	248 (28.4)		
High K10	208 (21.2)	25 (22.9)	183 (20.9)		
Very High K10	189 (19.2)	20 (18.4)	170 (19.5)		
Low/moderate K10	585 (59.5)	64 (58.7)	521 (59.6)	0.875	0.563
High/Very High K10	398 (40.5)	45 (41.3)	353 (40.4)		

Data are presented as mean (standard deviation) for continuous and number (percent) for categorical data. Data were analysed with an independent *t*-test for continuous normally distributed and a chi-squared test for categorical data. PCOS: polycystic ovary syndrome.

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
