# Peer review of "Impact of COVID-19 Restriction on Weight, Physical Activity, Diet and Psychological Distress on People with PCOS"

_nutrients, 2023, doi:10.3390/nu15112579_

Round 1

Reviewer 1 Report

The authors report the impact of COVID 19 restrictions on people with and without PCOS. 

It would be useful to compare these changes to a period of time prior to the pandemic. People with PCOS usually gain more weight regardless of restrictions. It is unclear these differences are due to the pandemic.

Is there information on participants' utilzation of healthcare services for weight management or mental health during this period?

What should be done differently for PCOS patients based on this data? What additional resources can be provided to this population?

Reviewer 2 Report

The authors focused on the impact of COVID-19 Restriction on people with PCOS. They investigated the impact the COVID-19 restrictions had on weight, physical activity, diet and psychological distress for Australians with PCOS.

(1) The authors are suggested to provide the details of data collection and preprocessing and draw necessary example figures to clearly show the key steps of the used method.

(2) Since a total of 1005 women responded to the survey, how to split them into different groups?

(3) The literature search is weak.  The intersection of COVID-19 restriction, AI-based disease diagnosis, and the related applications in  PCOS is suggested to append, such as:

[1] A Survey on Applications of Artificial Intelligence in Fighting Against COVID-19. ACM Computing Surveys

[2] Automatic Fetal Ultrasound Standard Plane Recognition Based on Deep Learning and IIoT.  IEEE TRANSACTIONS ON INDUSTRIAL INFORMATICS

[3] DeepR2cov: deep representation learning on heterogeneous drug networks to discover anti-inflammatory agents for COVID-19. Briefings in bioinformatics

(4) More discussion is suggested to support the conclusions that periods of COVID-19 lockdowns resulted in increased levels of stress, depression and anxiety.

(5) Proofreading is suggested.  Some grammar errors and improper phrases are found.

NA

Round 2

Reviewer 1 Report

The authors have addressed some of the concerns. Due to the study type it is understandable that utilization of healthcare during the pandemic cannot be assessed. The other changes are appropriate.

Author Response

Thank you for your review.